# Ensemble Siamese Network (ESN) Using ECG Signals for Human Authentication in Smart Healthcare System

**DOI:** 10.3390/s23104727

**Published:** 2023-05-13

**Authors:** Mehdi Hazratifard, Vibhav Agrawal, Fayez Gebali, Haytham Elmiligi, Mohammad Mamun

**Affiliations:** 1Department of Electrical and Computer Engineering, University of Victoria, Victoria, BC V8W 2Y2, Canada; 2National Research Council of Canada, Government of Canada, Ottawa, ON K1A 0R6, Canada

**Keywords:** Ensemble Siamese Network, dynamic authentication, IoT security, continuous authentication, deep learning, smart healthcare system

## Abstract

Advancements in digital communications that permit remote patient visits and condition monitoring can be attributed to a revolution in digital healthcare systems. Continuous authentication based on contextual information offers a number of advantages over traditional authentication, including the ability to estimate the likelihood that the users are who they claim to be on an ongoing basis over the course of an entire session, making it a much more effective security measure for proactively regulating authorized access to sensitive data. Current authentication models that rely on machine learning have their shortcomings, such as the difficulty in enrolling new users to the system or model training sensitivity to imbalanced datasets. To address these issues, we propose using ECG signals, which are easily accessible in digital healthcare systems, for authentication through an Ensemble Siamese Network (ESN) that can handle small changes in ECG signals. Adding preprocessing for feature extraction to this model can result in superior results. We trained this model on ECG-ID and PTB benchmark datasets, achieving 93.6% and 96.8% accuracy and 1.76% and 1.69% equal error rates, respectively. The combination of data availability, simplicity, and robustness makes it an ideal choice for smart healthcare and telehealth.

## 1. Introduction

The combination of healthcare services and technology is known as digital healthcare. It encompasses various technological tools like mobile health apps, electronic health records, wearable devices, and personalized medicine. This blend of technology and healthcare offers an opportunity to leverage data and technology to enhance patient health and care quality while also providing a secure platform [1]. The healthcare sector is now in a new phase of technological development, known as Industry 5.0, which is leading to innovative applications that can be categorized as intelligent healthcare. Some examples of such applications include using augmented reality for clinical decision support, IoT-based health prescription assistant, remote diagnosis, digital non-invasive medical techniques, collaborative robots for complex medical procedures, and cloud-based real-time prediction of patient status.

The healthcare sector’s digital transformation has increased its vulnerability to both cyber-attacks and healthcare system resilience. Healthcare resilience is described as “the ability of healthcare systems to succeed under varying conditions” [2]. That is, digital healthcare has been a longstanding cause of concern [3,4]. Besides the security of patient health information, privacy is also of significant concern as electronic health records contain patients’ sensitive data. Authentication is one of the most commonly adopted measures to allow authorized access. Biometric authentication systems are the most preferred, including an individual’s behavioral and physiological characteristics. It comes as no surprise that the biometric market worldwide is projected to surpass 50 billion USD by 2024, exhibiting a steady annual growth rate of 20% [5].

Behavioral biometrics, such as voice recognition, keystroke, and touch dynamics, gait, posture, etc., are often cumbersome and unreliable as they vary depending on mood, age, tiredness, etc. Physiological biometric authentication systems, which include fingerprint, iris recognition, facial recognition, etc., are being widely adopted as they remain stable throughout an individual’s lifetime [6]. The main advantage of biometric authentication, which cannot be neglected, is that it can be performed continuously and reliably with minimum action or intervention from the user. That is, a biometric authentication system can provide security along with usability. The biometric authentication system can be deployed in various applications, from real-time to commercial ones, and can even be used to access various services related to healthcare.

Physiological signals are considered the preferred method for authentication due to their difficulty in being counterfeited and the requirement for individuals to present themselves to record such signals [7]. The most prominent physiological signals include electrocardiogram (ECG), electroencephalography (EEG), and photoplethysmography (PPG) [6]. Obtaining PPG and ECG signals is relatively easy compared to most other physiological signals, with ECG signals being more resilient to noise compared to PPG or EEG signals, making ECG the preferred biometric system [8]. For example, simple finger sensors can be used to obtain PPG and ECG signals [9]. With the recent introduction of ECG sensors in commercial products such as wearables, the possibility of widespread adoption of ECG biometrics has increased [10].

Independent research proves the authenticity of ECG signals [8,11,12,13]. Guglielmi et al. [11] collected ECG signals from various sensor nodes on the individual’s body and used them to develop a new key agreement protocol that exploits the randomness of ECG signals. An analytical solution is provided to ensure robust authentication and a unique information reconciliation matrix is developed that offers good performance for all ECG sensor pairs. The approach offers evidence to support the authenticity of ECG signals for authentication purposes. ECG signals can still be effective for authentication even when the signal is affected by unexpected problems [13]. Prakash et al. investigated the effectiveness of using ECG signals for authentication in scenarios where the signal is affected by uncontrolled acquisition conditions. They found that ECG signals can still be used for authentication purposes even when the signal is distorted by various factors such as motion artifacts, electrode detachment, or changes in heart rate. Pereira et al. [14] investigated the performance of ECG-based authentication when the signal is affected by sudden cardiac events. The authors found that despite the limited data availability caused by sudden cardiac events, ECG-based authentication remains effective. They concluded that ECG-based authentication could provide a reliable and secure authentication method even when the signal is affected by unexpected problems.

In order to achieve practical and real-time authentication based on collected ECG signals, machine learning (ML) methods must be employed. Kashou et al. compared the ability of ML and deep neural networks (DNNs) using ECG signals for authentication [12]. Developing a verification model for user identification using ML and DLLs offers many advantages, such as being human-independent, cost-effective, reliable, faster, and more precise [15]. Numerous ML models have been developed for biometric-based authentication systems, with DNNs being widely applied in this area recently [16]. DNNs are preferred because they can extract the best feature set from raw data during training without requiring extra effort. Despite many advantages of DNN models being used for authentication, they have some shortcomings, which are outlined below:Requiring a large amount of data to train the model [17].Most ML models are sensitive to imbalanced datasets [18].Enrolling a new entity in the system is complex because of the need to train the model from scratch [15].

The Siamese network is a type of deep neural network that can tackle the problems mentioned above. It comprises identical subnetwork components that are used in various studies, including facial and palmprint recognition [19,20]. The main idea behind the first part of the Siamese network is to extract the best features to compare the inputs of the separate sub-networks. The second stage of the Siamese network is responsible for assessing the similarity between input sets. For this purpose, pairs of samples from both the same person and different individuals are used as training samples. Since we can select these couple of samples from all classes in the dataset, it is not a big deal to make many couples from available samples and overcome the small sample size problem [21]. Furthermore, selecting couples can alleviate the imbalance condition and make this model insensitive to an imbalanced initial dataset. ML algorithms generally exhibit good performance on balanced datasets, but when faced with imbalanced datasets, their performance may suffer [22]. Another advantage of the Siamese network is that it is not necessary to use the enrolled persons for model training. For adding a new user to the system, it just needs to add their registered features to the repository.

We used the Siamese network in our previous research paper [23] to perform authentication using ECG signals. The current study improves the ability to deal with ECG signals in diverse situations. Utilizing an Ensemble network can help the authentication system compare a new instance from a user to many previous records from them instead of only one. We summarize this study by the following key contributions:Proposing an Ensemble Siamese network (ESN) using ECG signals to compare a new sample with the saved records from the claimed identity in the repository.Saving encoded data from registered users in a repository instead of having the raw information to increase data safety.Using a modified Fourier transform customized for this application to extract more reliable features from the input samples, which results in training a more robust network.Developing a strong authentication method that can enroll new users at run time while being robust to imbalance sample conditions.

The remainder of this paper is structured as follows: Section 2 explores related work and examines significant research studies in this field. The proposed ensemble model is presented in Section 3. Subsequently, Section 4 outlines the dataset used to evaluate the proposed model and compares the results of our implementation with state-of-the-art methods. In Section 5, we conclude our work and suggest future improvements for this method.

## 2. Related Work

The ECG signal is a widely researched and recognized medical signal that depicts the electrical activity of muscle fibers in various regions of the heart. It serves the purpose of identifying patients, as well as real-time monitoring and detecting numerous cardiovascular system-related medical issues.

In recent years, ECG data has become increasingly relevant for user authentication and identification due to its unique characteristics related to an individual’s physiology. Unlike other biometric traits, such as fingerprints, face, voice, or retina, the ECG signal is difficult to replicate without direct access to the user and is not affected by lighting conditions or background noise, making it highly reliable. The ECG signal can be obtained from a variety of devices, making it convenient for multimodal systems that combine ECG with other biometrics to increase security [24]. The ECG also provides inherent liveness detection, as a deceased individual no longer expresses this feature. However, using the ECG as a biometric has some drawbacks, including the fact that it cannot be replaced if it falls into the wrong hands and that the signal may contain sensitive information regarding the user’s health conditions, wellness, or mood. As such, collecting, processing, storing, and transmitting data carefully is essential if using the ECG as a biometric. Nevertheless, when used correctly, the ECG can be a robust user authentication and identification system option [11,12,13,25].

Recent studies have shown that ECG signals can be used to improve the security of wearable devices and internet of things (IoT) applications. For example, researchers have developed a wearable device that uses ECG data for user authentication and identification, showing that it can achieve high accuracy and reliability [26]. Furthermore, studies have also investigated the use of ECG data in combination with other biometric modalities, such as facial recognition, to enhance the security of biometric systems. These studies have shown that combining ECG with other biometrics can improve both accuracy and robustness, making it an attractive option for future biometric systems.

The ECG is a reliable medical signal that has long been used to record the electrical activity of heart muscles in various regions of the heart, enabling the detection of different heart conditions and abnormalities. However, it has also emerged as a surprising but viable option for telehealth authentication systems. This is due to two main reasons. Firstly, ECG features are easily accessible through most digital healthcare systems and smart mobile devices, making it a widely available option. Secondly, ECG is unique to each individual, making it an excellent choice for identification purposes. ECG-based authentication systems are categorized into two types based on the features used for recognition: fiducial and non-fiducial [27]. Fiducial methods extract six characteristic points (P, Q, R, S, T, U) from the heartbeat wave and utilize them to derive features such as amplitude and latency. Non-fiducial techniques, on the other hand, analyze either the entire ECG waveform or a portion of it to extract valuable information. Partially fiducial methods combine the two approaches [28].

Numerous DNN architectures have been suggested for authentication systems based on ECG. Among them, the convolutional neural network (CNN) is widely used for analyzing data with stationary features, which are concepts that are repeated in the samples [29]. CNN-based architectures for classification and authentication mainly consist of three layers: convolution, pooling, and fully connected. The first two layers are utilized for extracting features from inputs, while the last layer utilizes the extracted features to achieve the objective, such as classification [30]. In sequence prediction problems, classifiers take an input instance and assign it to one of the available classes. Long short-term memory (LSTM) is a type of recurrent neural network (RNN) that has found extensive use in applications such as speech recognition [31], text classification [32], and ECG biometrics [33]. In this field, a hybrid model called CNN-LSTM is utilized. This model combines a CNN with an LSTM architecture, specifically created for sequence prediction problems that involve spatial inputs, such as images or videos [34]. These models aim to extract more useful features from raw data by taking into account recurring patterns over time.

Earlier research papers have reported that the CNN, LSTM, and CNN-LSTM architectures for ECG authentication were able to achieve a classification accuracy of over 90% [35]. Although none of the proposed models offer a direct authentication approach that can be implemented without post-processing after classification, supervised DNN-based models, like classifiers, are commonly employed for authentication [36]. The features extracted prior to the last layer of DNN models are typically stored in a database. During the testing phase, each input is compared to the previous features stored in the database using a predetermined threshold. Although this method has been used for authentication purposes, the final authentication model’s performance is often not as accurate as classification models [15]. Autoencoders are deep neural networks that can encode data in an efficient manner, and then reconstruct the original data from the compressed representation without any supervision [37]. One significant disadvantage of these models is that they require training a separate model for each user in the system.

To overcome this issue and enable one-shot authentication, the Siamese network has emerged as a promising solution. Several research papers have investigated the use of Siamese networks for ECG-based authentication systems. Ibtehaz et al. [6] reported achieving 99% classification accuracy using a single heartbeat and 100% accuracy by combining multiple beats. They also proposed a direct authentication model using the Siamese architecture that achieved 88.3% accuracy and an equal error rate (EER) of 2.29%. Ivanciu et al. [38] introduced an innovative Siamese network structure that used images as input for authentication, which was a departure from the traditional use of signals. Although the model encountered challenges in processing and eliminating noise from the input data, it achieved an overall accuracy of 86.4% in authentication and an EER of 2.95%.

There are other preprocessing methods that aim to remove noise and reduce the initial signal dimensionality, but this can result in a loss of small information before classification. Rabcan et al. [39] proposed an approach for EEG signal classification that considers this loss of information by using a fuzzy classifier. They used a Fuzzy Decision Tree (FDT) as the classifier, which takes the uncertainty of initial data into account caused by loss of information during dimensionality reduction. However, applying the fuzzy classifier requires modifying the preprocessing step to include a fuzzification procedure. While the approach and processing methods used for the analysis and classification of ECG and EEG signals may differ due to their inherent differences in signal characteristics, this is a trend that can be inspired for processing ECG signals.

## 3. The Proposed Method

The Siamese network was utilized in our prior study [23] to tackle challenges related to limited sample availability and imbalanced scenarios in the context of ECG-based authentication. A major concern with all available methods is that they have been evaluated only on standard data collected on lab conditions. While if the signal conditions change regarding the situation, they may not perform well in real problems. The main advantage of the proposed model is utilizing an ensemble Siamese network to extract features from different signals collected from various situations of a user, which enables the model to tolerate different signals from a person. Figure 1 illustrates the schematic view of our model, which consists of four primary steps: data collection, Preprocessing and Data Preparation, Siamese network training, and the test phase.

### 3.1. Data Collection

The first step is accessing data to be used in the corresponding ML lifecycle [15]. There are several ways to collect ECG signals for user authentication in a digital healthcare system. Here are some common methods:Wearable devices: ECG sensors can be integrated into wearable devices such as smartwatches, wristbands, or patches.Electrodes: ECG electrodes can be attached to the user’s chest, arms, or legs to collect the ECG signal.Smartphones: Some smartphones have built-in ECG sensors that can collect the user’s ECG signal.Holter monitors: Holter monitors are portable ECG machines that can continuously monitor the user’s ECG signal for up to 24 h.

These devices can continuously monitor the user’s ECG signal and transmit the data to the healthcare system for authentication purposes.

### 3.2. Preprocessing and Data Preparation

ECG signals can be used for authentication in two ways:Using machine learning models on raw ECG signals: In this approach, the raw ECG signal is used as input to a machine learning model such as a neural network or a support vector machine.Using Fourier transform and then machine learning: In this approach, the ECG signal is first transformed into the frequency domain using Fourier transform. The resulting frequency spectrum is then used as input to a machine learning model.

**Figure 1 sensors-23-04727-f001:**
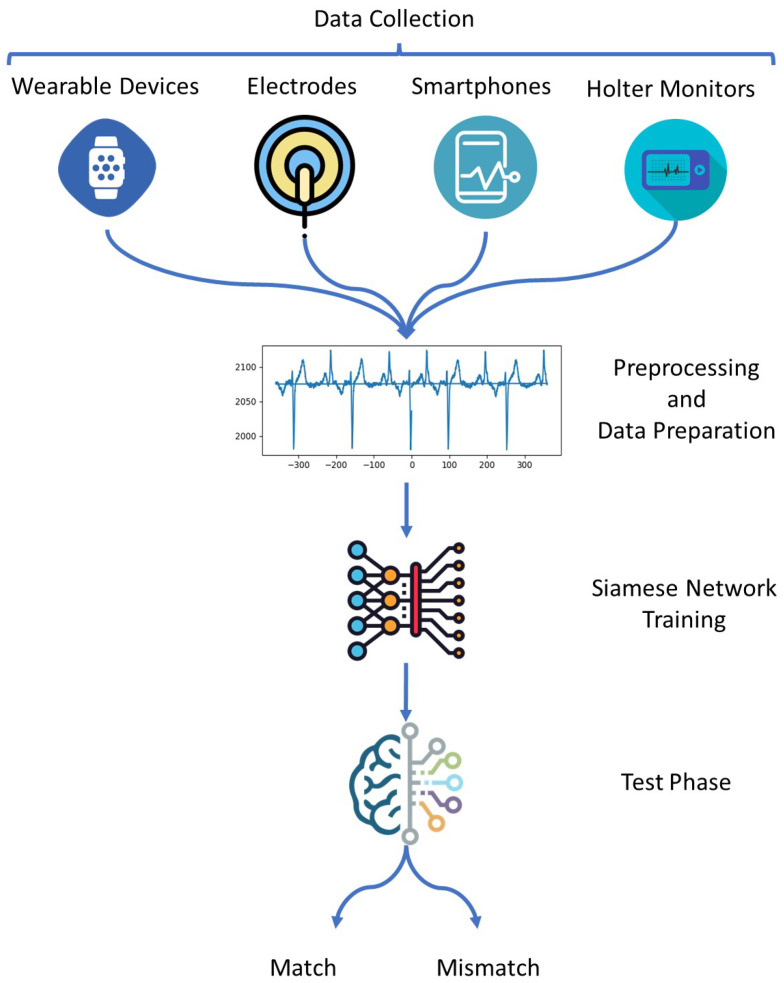
The overall view of the proposed method from data collection to detecting match/mismatch samples.

Both approaches have their advantages and limitations. Using machine learning models on raw ECG signals can be more accurate in identifying unique patterns in the signal. However, the model may require a large amount of training data, and it may be sensitive to noise and other signal artifacts. On the other hand, using Fourier transform and then machine learning can be more robust to noise and signal artifacts. Considering the lack of samples in authentication systems, We utilize the Fourier transform on input signals to extract dependable features from samples for network training. The discrete Fourier transform (DFT) is a practical tool for studying the ECG signal after it is digitized. The ECG signal has been described in the equation:(1)$$x\left[n\right]=\frac{1}{N}\sum_{k=-N/2}^{N/2-1}X\left[k\right]\;e^{j2\pi kn/N}$$
where *N* is the number of samples of the ECG signal and the time index is defined in the range 0≤n<N−1 and frequency index is defined in the range N/2≤k<N/2−1.

The Fourier complex coefficients X[k] are given by
(2)$$X\left[k\right]=\sum_{n=0}^{N-1}x\left[n\right]\;e^{-j2\pi kn/N}$$

The mathematical technique used in our paper is the Fast Fourier Transform (FFT), which converts a time-based function to a frequency-based function. The FFT is a version of the Discrete Fourier Transform that is efficiently implemented using mathematical methods. We use the FFT algorithm to extract feature points from ECG signals, such as PQRST points [6,40].

To prepare input data to train our model, we need to select *n* samples from each class. The (n+1)th signal is the new signal. If the new signal is from the same class as the first *n* samples, we label it as a positive result, and if it is from a different class, its label is negative. One of the advantages of using the Siamese network is that regardless of the number of samples from each class in the initial dataset, we can create balanced positive and negative bundles of samples in this phase and avoid the imbalance condition to train the model.

### 3.3. Training the Siamese Network

Figure 2 shows the overall view of the Siamese network we used in our proposed model. The network is trained using n+1 preprocessed samples as inputs. The CNN networks are responsible for extracting the optimal feature vector, which will be compared in the next section. The output of the comparison section is the similarity value between the feature vectors extracted from the input signals. To determine the similarity between the two inputs, we used the inverse of the $L_{1}$-norm and the Manhattan distance. A normal sigmoid function with a threshold of 0.7 was used in the next level to distinguish accepted and unaccepted tuples. The output is a value between 0 and 1, with a result close to 1 indicating that the test sample and saved records belong to the same person and a result close to 0 indicating that the input signals are from two non-identical individuals.

As depicted in Figure 2, both CNN networks have the same architecture, and the parameters are shared between them. The max pooling layer is used to calculate the most significant value in each feature map region, leading to a reduction in the number of dimensions. These networks aim to extract the feature vectors from the signals. Then, the absolute differences between the two vectors are calculated to compare them. Finally, the Sigmoid layer is applied to obtain the similarity value.

### 3.4. Test Phase

To register a user in the proposed authentication system, *n* sample records are required to be stored in a repository. Furthermore, we employ a trained Siamese network to establish a database for registered users. Each signal that enters the system to perform a test should have the same preprocessing with the shared parameters to have a unique structure in the model. After applying the preprocessing step, the extracted vectors by the CNN part are saved for future comparison. Thus, retraining the model is unnecessary while registering a new user. The extracted features from the sample records of the new person from the trained CNN section are adequate for storing. Consequently, there is no need to retain all CNN inputs in the test model. Saving one of them is sufficient since all branches share the same parameters. The overall structure of the final trained model is illustrated in Figure 3. The new input is passed through that single path, and the extracted vector is compared with the coded data in the repository from the same person. Finally, based on the similarity value, the model can accept or reject the input sample.

## 4. Experimental Results

This section will cover the experimental setup and results of implementing the proposed approach on two standard datasets. First, we will introduce the datasets and metrics needed to evaluate the performance of the final model. We will also analyze the hyperparameters used in the final model and compare the implementation results with other models to provide a comprehensive evaluation.

### 4.1. Dataset

ECG-ID and PTB are two widely used datasets for ECG signal analysis and authentication. While both datasets contain ECG signals, they differ in their sample size, source, and purpose. ECG-ID consists of ECG signals from 310 subjects recorded in controlled laboratory conditions from 90 individuals, including 44 men and 46 women. The signals were collected using a standard 12-lead ECG system and are relatively clean and noise-free. The purpose of this dataset is to study biometric identification using ECG signals. On the other hand, PTB is a dataset of ECG signals from 549 subjects with various cardiac conditions recorded in clinical settings from 290 signals. The signals were collected using a standard 12-lead ECG system and may contain noise and signal artifacts. The purpose of this dataset is to aid in the diagnosis and management of cardiac conditions.

Each ECG signal was digitized at 500 Hz with 12-bit resolution over a nominal range of ±10 mV, while each PTB signal was digitized at 1000 samples per second with 16-bit resolution over a range of ±16.384 mV. In terms of sample size, PTB has a larger number of subjects and signals compared to ECG-ID. However, ECG-ID has a more controlled and standardized environment for signal collection compared to PTB, which was collected in clinical settings. The noise and artifacts present in the PTB signals may affect the performance of the authentication system. The characteristics of these datasets are summarized in Table 1.

While both ECG-ID and PTB datasets contain ECG signals, they differ in sample size, source, and purpose. The choice of dataset depends on the specific requirements and application of the authentication system. If a controlled and clean environment is necessary, ECG-ID may be preferred, while if a more diverse and real-world dataset is required, PTB may be more suitable.

We evaluated Our proposed model using these two benchmark datasets. To this end, we utilized signals to train the Siamese Network. Samples of the ECG-ID and PTB signals are shown in Figure 4 (left and right, respectively). Furthermore, we examined the hyperparameters used in the final model and compared and discussed the implementation results with rival models.

### 4.2. Evaluation Metrics

During training, the cross-entropy loss is used as the evaluation metric for adjusting the model weights. As the output of the model is a probability or a similarity score between 0 and 1, a sigmoid function is employed in the last step to map the output to either 0 or 1. The accuracy metric is utilized to assess how many data samples are accurately authenticated or rejected by our model.

In the realm of authentication, the evaluation of a biometric model is determined by three error parameters: false acceptance rate (FAR), false reject rate (FRR), and equal error rate (EER), to provide a more specific metric. FAR and FRR indicate the percentage of false users authorized and the percentage of legitimate users rejected by the model, respectively. EER refers to the threshold values for FAR and FRR and indicates the point at which the FAR is equal to the FRR. A lower EER indicates a more accurate biometric system. Figure 5 illustrates the EER for two biometric systems [41]. The system depicted by the solid lines in this diagram exhibits superior performance compared to the one represented by the dashed lines.

### 4.3. Hyper Parameters and Experimental Results

After applying the Sines function for pretraining, the explained FFT is implemented on both datasets in our proposed model. We utilize the Fourier series on the PTB dataset, which changes it from 200 × 1 to 200 × 2. Similarly, the ECG-Identification (ECG-ID) dataset is transformed from 256 × 1 to 256 × 2.

The preprocessing phase involves feeding each signal into a CNN model consisting of eight 1D convolutional layers with ReLU activation functions. The number of filters varies across the layers, with 32 filters for the first two, 64 for the next two, 128 for the following two, and 256 for the last two layers. The kernel size is set to 3, and a Max-pooling layer is used after each convolutional layer with the same padding. The output of these layers is then flattened, and a dense layer of 512 neurons is applied. The last layer of the model uses a sigmoid activation function to compute the Euclidean distance between four signals, three from the same person and one to be compared with them (from the same or a different person). To prevent overfitting and constraining weight growth, $L_{2}$ regularization is applied. The model uses the binary cross-entropy loss function and the Adam optimizer.

In the preprocessing phase, we extracted 2000 sets of three samples from the same persons. In the next step, we added a sample from the same person of each set to 1000 sets and a sample from different persons to the other 1000 sets. This way, we create 1000 sets with the label of positive and 1000 sets with the label of negatives and address the imbalance conditions in the initial dataset. To be fair, we exerted 10-fold cross-validation in splitting created dataset to train and test.

Table 2 compares the results of ESN with the rival methods on the ECG-ID dataset. Our model shows 93.6% average accuracy, 1.69% FAR, 1.84% FRR, and 1.76% EER, which outperforms the rivals. The rival models include the proposed models in [6,23,38] that used ECG signals in the Siamese network for authentication in their paper. Also, to find the best structure in the proposed model, we used CNN-LSTM, LSTM, and Bi-LSTM structures in a similar Siamese network using the same preprocessing as our proposed model.

ECG-ID samples are collected in a controlled and clean environment, and we performed this test just to compare our model in laboratory conditions with the rival methods. This comparison aims to find the best structure for feature extraction from the ECG signals. Experimental evaluation proves that CNN is the best structure for feature extraction from ECG signals in the Siamese network.

Table 3 presents the results of the proposed model on the PTB dataset. Our model shows 96.8% average accuracy, 1.69% FAR, 1.73% FRR, and 1.66% EER on the PTB dataset. Again the rival models include the proposed models in [6,23,38] and also using CNN-LSTM, LSTM, and Bi-LSTM structures with the same preprocessing as our proposed model.

PTB is a more diverse and real-world dataset that can challenge our model and extend it to an actual application. The original purpose of this dataset is to aid in diagnosing and managing cardiac conditions. As a result, it is noteworthy to test our model under different conditions of ECG signal collection for the patients. Comparing the results of the proposed model with state-of-the-art models shows that the proposed model outperforms the rivals.

## 5. Conclusions and Future Work

Traditional authentication methods typically classify samples, whereas our proposed model directly performs authentication. To this end, we introduced an Ensemble Siamese Network (ESN) model for ECG signal authentication. ECG signals are readily accessible in telehealth, and many modern smartphones are equipped with sensors to collect such data, making our proposed model suitable for use in the digital healthcare environment. The ESN architecture employs a CNN model to process the input signals, and the comparison section of the network compares the corresponding extracted features with the claimed sample vectors in the repository to calculate their similarity.

Using an ESN for ECG signal authentication can help consider different user conditions while collecting ECG signals. An ESN compares a new signal with many available samples to make predictions, which can improve the accuracy and robustness of the model. Therefore, the system can account for individual differences in ECG signals that may arise due to different health conditions, physical activity levels, or other factors that affect the signal. This can improve the accuracy and reliability of the authentication system, making it more suitable for use in real-world scenarios where users may have different conditions or circumstances. Additionally, using an ESN can help to address issues such as imbalanced datasets or difficulty in enrolling new users, which are common challenges in authentication models that rely on machine learning.

The experimental evaluation of our model was conducted on two benchmark datasets; ECG-ID and PTB. The ECG-ID dataset was collected in a controlled and clean environment, while the PTB dataset had a more diverse and real-world condition. The experimental results show that the proposed model outperforms the rival methods in both states. Also, using CNN structure performs better than other available structures, such as LSTM, CNN-LSTM, and Bi-LSTM, in the feature extraction part of the model.

Here are the main advantages of the proposed ESN model:Utilizing ECG signals in a Siamese Network platform that can be trained using a few samples.The robustness of the model using a CNN for feature extraction and the capability to handle imbalanced datasets are strengths of our approach.Utilizing an ensemble of some samples that can tolerate changes in signal conditions and increase the model robustness.Directly registering new users in the system by encoding their data using the trained CNN layers and storing them in a repository. This way, feature extraction on saved records is not required for online applications, which leads to a decrease in the processing time. Furthermore, data vulnerability can be avoided by accessing the unknown entity.

We suggest extending the ensemble method to a diverse range of collected features from individuals to continue this research trend. For example, we can collect data according to typing patterns or keystroke dynamics of the system’s user while simultaneously collecting their face image and ECG signal to increase the model’s stability and robustness. This suggestion can raise the cost of impersonation, and it will be almost impossible to impersonate a person with various behavioral features. On the other hand, adding other techniques, such as fuzzification, may help to overcome noisy data challenges.

## Figures and Tables

**Figure 2 sensors-23-04727-f002:**
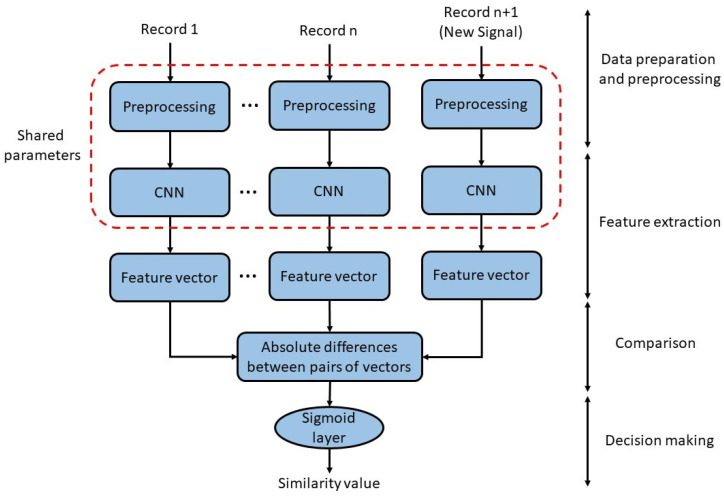
The proposed method involves three main components: preprocessing, feature extraction, and comparison. At the first stage of our proposed model, input signals are fed into distinct branches of the Siamese network. The signals go through identical preprocessing and feature extraction processes, and then their comparison takes place in the subsequent stage.

**Figure 3 sensors-23-04727-f003:**
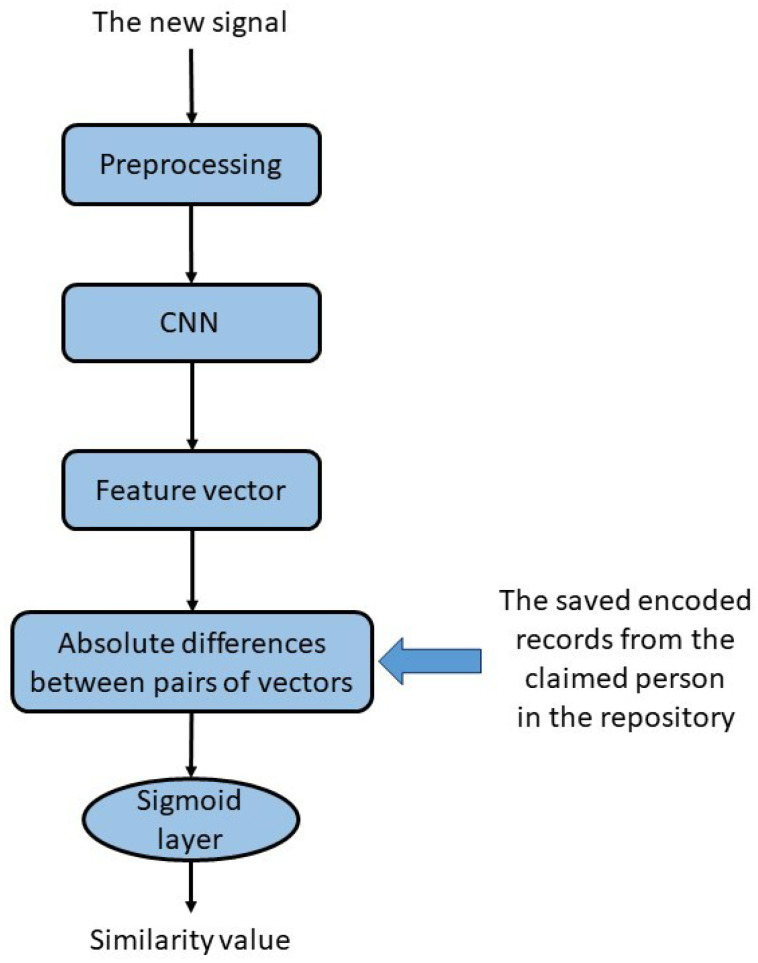
The final trained model can be illustrated through a schematic view. The model requires only one branch to encode the new input sample. The encoded representation is then compared with the saved encoded records of the claimed person stored in the repository.

**Figure 4 sensors-23-04727-f004:**
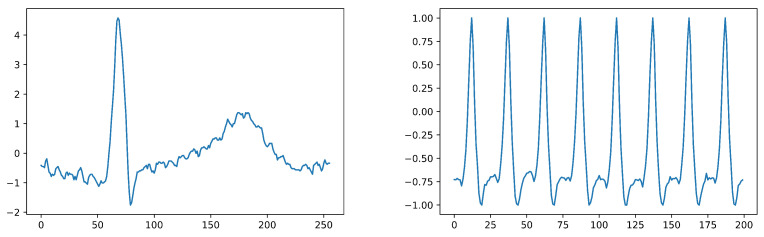
The image on the **left** shows a sample of an ECG-ID signal, while the image on the **right** displays a sample of a PTB signal.

**Figure 5 sensors-23-04727-f005:**
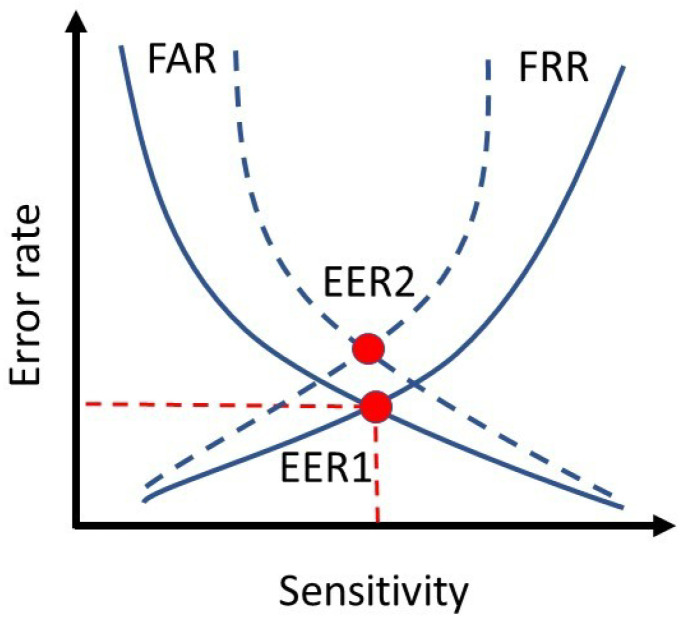
Comparing EER for two different biometric systems. The EER diagram with solid lines got a better result than the other one with dashed lines.

**Table 1 sensors-23-04727-t001:** Details of the benchmark datasets, which consist of ECG signals for the purpose of authentication, are presented in this table.

Database	# of Records	# of Subjects	Resolution	The Nominal Range
ECG-ID	310	90 (44 men and 46 women)	12-bit	±10 mV
PTB	549	290 (209 men and 81 women)	16-bit	±16.384 mV

**Table 2 sensors-23-04727-t002:** This table compares the accuracy, FAR, FRR, and EER percentage of the proposed model with other existing methods on the ECG-ID dataset for authentication.

Method	Accuracy (%)	FAR (%)	FRR (%)	EER (%)
The proposed model (ESN)	93.6	1.69	1.84	1.76
Behrouzi et al. [23]	92.1	1.91	1.99	1.94
Ibtehaz et al. [6]	88.3	2.36	2.20	2.29
Ivanciu et al. [38]	86.4	2.77	3.06	2.95
CNN-LSTM	86.8	2.16	2.11	2.13
LSTM	91.4	2.04	2.12	2.09
Bi-LSTM	89.1	2.04	2.17	2.07

**Table 3 sensors-23-04727-t003:** This table compares the accuracy, FAR, FRR, and EER percentage of the proposed model with the rival methods on the ECG-ID dataset for authentication.

Method	Accuracy (%)	FAR (%)	FRR (%)	EER (%)
The proposed model (ESN)	96.8	1.73	1.66	1.69
Behrouzi et al. [23]	95.3	1.89	2.01	1.94
Ibtehaz et al. [6]	90.4	1.99	1.93	1.97
Ivanciu et al. [38]	92.3	2.20	2.39	2.27
CNN-LSTM	91.7	2.11	2.01	2.06
LSTM	92.3	2.03	1.87	1.92
Bi-LSTM	90.9	1.86	2.05	1.98

## Data Availability

Here are the links to two datasets that we used to evaluate our model and compare it to the state-of-the-art: https://physionet.org/content/ecgiddb/1.0.0/, https://www.physionet.org/content/ptbdb/1.0.0/ (accessed on 1 March 2023).

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
