# Peer review of "Ensemble Siamese Network (ESN) Using ECG Signals for Human Authentication in Smart Healthcare System"

_sensors, 2023, doi:10.3390/s23104727_

Round 1
Reviewer 1 Report
In this paper, the problem of low sensitivity of the authentication model of machine learning to imbalanced data sets is studied, and ECG signals that are easy to access in the use of digital medical systems are proposed, through an integrated Siamese network (ESN) that can handle small changes in ECG signals ) for authentication. After experiments and verification, the extracted results show that compared with the state-of-the-art, the model trained on the ECG-ID and PTB benchmark datasets achieves 93.6% and 96.8% accuracy and 1.76% and 1.69% accuracy respectively. Equal error rate. Data availability, simplicity, and robustness have been significantly improved. I think this study has certain significance. After reviewing the manuscript, I would like to make suggestions for some problems in the article I found:
1. It is recommended to add more experimental parts to ensure the accuracy of the article;
2. The flow chart of this article is too monotonous and it is recommended to modify it;
3.I understand that a lot of work has been done on researching related problems, such as [A]-[C]. Authors are advised to cite and discuss recent literature in this field.
A. DOI:10.1109/JPHOT.2022.3233129
B. DOI:10.1016/j.eswa.2023.119591
C. DOI:10.1016/j.jelectrocard.2023.03.009
In addition, I think there are still many basic errors to be solved in the paper, such as some basic format errors that need to be paid attention to:
A. The use of fig and figure needs to be unified;
B.It is recommended to use vector graphics for articles;
C.The picture suggestion in Figure 4 shows what it means.
Based on the above report, my review result is a minor revision.
Author Response
Response to Reviewer 1 Comments
Ensemble Siamese Network (ESN) Using ECG Signals for Human Authentication in Smart Healthcare System
Dear Editor,
We would thank the reviewers for their valuable comments, which helped improve the manuscript.
Below are our answers to the reviewers’ comments and the modifications made to accommodate
these comments. (You can find the addressed modifications in red in the updated paper.)
Best regards,
Seyed Mehdi Hazratifard, Vibhav Agrawal, Fayez Gebali, Haytham Elmiligi, and Mohamad Mamun
Point 1: It is recommended to add more experimental parts to ensure the accuracy of the article;
Response 1: To address your comment, we added FAR and FRR percentages to evaluate our model and compare it with the rival methods (page 11, tables 2 and 3). Also, we added two investigation paragraphs to interpret each one (page 10, the last paragraph, and page 11, the first paragraph).We compared the proposed model with the rivals using two datasets in controlled and uncontrolled conditions and discussed the results.
Point 2: The flow chart of this article is too monotonous and it is recommended to modify it;
Response 2: To address your comment, we added Figure 1, which is a flowchart with more details on page 5.
Point 3: I understand that a lot of work has been done on researching related problems, such as [A]-[C]. Authors are advised to cite and discuss recent literature in this field.
- DOI:10.1109/JPHOT.2022.3233129
- DOI:10.1016/j.eswa.2023.119591
- DOI:10.1016/j.jelectrocard.2023.03.009
Response 3: Thanks for your advice. We added the related reference to investigate ECG signals’ nature and their applications using machine learning and deep learning algorithms for authentication. (page 2, paragraph 3)
Point 4: In addition, I think there are still many basic errors to be solved in the paper, such as some basic format errors that need to be paid attention to:
- The use of fig and figure needs to be unified;
B.It is recommended to use vector graphics for articles;
- The picture suggestion in Figure 4 shows what it means.
Response 4: Thanks for your good point; we unified all “fig.” into “Figure.”
We see what you meant and substituted other pictures with high-quality ones.
Based on the above report, my review result is a minor revision.
We would like to appreciate your valuable and productive comments.

Reviewer 2 Report
The authors attempted to improve their previous work regarding Human Authentication in Smart Healthcare System with Siamese network. In general, the topic and scope of the paper seem okey. However, when looking at their previous work, there are some serious issues that must be considered. For example:
-The same Siamese network was used in previous paper.
-The same dataset was used.
-The same performance metrics were used.
-The results look very similar.
-A major problem is that many figures are exactly same in both paper which is not scientific way.
-I see another serious problem about the results. Both papers use the same dataset and network but the performance outputs vary for compared schemes. For example, why CNN-LSTM has different performance accuracy under same settings?
Therefore, there is a strong need to clarify the differences between the previous and present studies.
Author Response
Response to Reviewer 2 Comments
Ensemble Siamese Network (ESN) Using ECG Signals for Human Authentication in Smart Healthcare System
Dear Editor,
We would thank the reviewers for their valuable comments, which helped improve the manuscript.
Below are our answers to the reviewers’ comments and the modifications made to accommodate
these comments. (You can find the addressed modifications in red in the updated paper.)
Best regards,
Seyed Mehdi Hazratifard, Vibhav Agrawal, Fayez Gebali, Haytham Elmiligi, and Mohamad Mamun
The authors attempted to improve their previous work regarding Human Authentication in Smart Healthcare System with Siamese network. In general, the topic and scope of the paper seem okey. However, when looking at their previous work, there are some serious issues that must be considered. For example:
Point 1: The same Siamese network was used in previous paper.
Response 1: We added some explanations to the paper to elaborate on the differences between the current work and the previous one (page 3, paragraph 2, and page 5, paragraph 3). Also, you can find this work’s contributions on page 3, paragraph 3. The main contribution of this work is proposing an Ensemble Siamese network (ESN) using ECG signals to compare a new sample with the saved records from the claimed identity in the repository. Comparing a new sample with more than one sample makes the model more robust and resilient (page 5, paragraph 3). Furthermore, we added a flowchart to show the overall view of the proposed method and its difference from the previous work (page 1, figure 1)
Point 2: The same dataset was used.
Response 2: These datasets have been used to assess many papers in this realm. We compared the proposed model with the rival methods using two datasets in controlled and uncontrolled conditions and brought the results on page 11, tables 2 and 3. Also, we added two investigation paragraphs to interpret each one (page 10, the last paragraph, and page 11, the first paragraph).
Point 3:The same performance metrics were used.
Response 3: To address your comment, we added FAR and FRR percentages to evaluate our model and compare it with the rival methods (page 11, tables 2 and 3). We compared the proposed model with the rivals using two datasets in controlled and uncontrolled conditions, discussed the results, and added two paragraphs to investigate and interpret the results (page 10, the last paragraph, and page 11, the first paragraph).
Point 4: The results look very similar.
Response 4: Using an ensemble method could improve the result a bit. However, the main objective of using an ensemble network in our proposed model was to compare each sample with a variety of samples to make the model more robust (page 5, paragraph 3). To verify our claim and prove our model’s robustness, we used the PTB dataset that is collected in an uncontrolled environment in differential diagnosis and management of cardiac conditions (page 11, paragraph 1)
Point 5: A major problem is that many figures are exactly same in both paper which is not scientific way.
Response 5: Thank you for the valuable comment. We eliminated the signal figures that were the same as the previous paper and added two different signals (page 9, figure 4). Furthermore, we added a flowchart to show the overall view of the proposed method and its difference from the previous work (page 1, figure 1)
Point 6: I see another serious problem about the results. Both papers use the same dataset and network but the performance outputs vary for compared schemes. For example, why CNN-LSTM has different performance accuracy under same settings?
Response 6: In this paper, we trained CNN-LSTM, LSTM, and Bi-LSTM models using the same input as our proposed model. As a result, the preprocessing that we applied to datasets caused changes in their results. We elaborated on this matter in the preprocessing subsection of the proposed method (page 6, paragraph 4).
Therefore, there is a strong need to clarify the differences between the previous and present studies.
Thank you for the valuable comments. We did our best to address all of them.

Reviewer 3 Report
The authors propose an original authentication method based on ECG signals, The method is original, but I doubt its practical suitability. I have some comments:
- How will authentication take place if, at the time of authentication, the patient encounters unexpected problems that affect the ECG signal? To what extent is the authenticity of the ECG signal proven?
- Could you introduce studies that prove the authenticity of the ECG signal in addition to [8]?
- Could you explain how will ECG and PTB signals be taken? Could you in more details describe the EEG signals and PTB signals?
- Could you explain in more detail the choice of classifiers for the study?
- Is the result stable depending on the initial data? Do you expect the result to change if others samples of signals are used?
- Could you do a preliminary evaluation for the authentication based on the application of a fuzzy classifier (https://ieeexplore.ieee.org/document/9224666 or )? Can be result be improved?
- Could you explain the principal steps of methods for signal classification as feature extraction, feature selection and classification in your study and the algorithms which are used for their implementation?
- Why do you use only the “accuracy” for the result evaluation? Could you add others metrics for classifier efficiency evaluation?
- Could you compare this authentication method with others?
Author Response
Response to Reviewer 3 Comments
Ensemble Siamese Network (ESN) Using ECG Signals for Human Authentication in Smart Healthcare System
Dear Editor,
We would thank the reviewers for their valuable comments, which helped improve the manuscript.
Below are our answers to the reviewers’ comments and the modifications made to accommodate
these comments. (You can find the addressed modifications in red in the updated paper.)
Best regards,
Seyed Mehdi Hazratifard, Vibhav Agrawal, Fayez Gebali, Haytham Elmiligi, and Mohamad Mamun
Point 1: How will authentication take place if, at the time of authentication, the patient encounters unexpected problems that affect the ECG signal? To what extent is the authenticity of the ECG signal proven?
Response 1: There are some studies that suggest ECG-based authentication can still be effective even when the signal is affected by unexpected problems. To address your comment, we added some of these studies to our paper. (page 2, paragraph 3)
Also, the main objective of using an ensemble network in our proposed model was to compare each sample with a variety of samples to make the model more robust. (page 5, paragraph 3)
To verify our claim and prove our model’s robustness, we used the PTB dataset that is collected in an uncontrolled environment in differential diagnosis and management of cardiac conditions (page 11, paragraph 1)
Point 2: Could you introduce studies that prove the authenticity of the ECG signal in addition to [8]?
Response 2: We added three references to prove the authenticity of ECG signals from two perspectives; one from ECG signals’ nature and the other from their applications using machine learning and deep learning algorithms. (page 2, paragraphs 3 and 4)
Point 3: Could you explain how will ECG and PTB signals be taken? Could you in more details describe the EEG signals and PTB signals?
Response 3: We described the datasets’ characteristics and their applications in more detail and added three paragraphs at the end of page 8 and the beginning of page 9.
Point 4: Could you explain in more detail the choice of classifiers for the study?
Response 4: We discussed the advantages of choosing the Siamese network as a useful classifier for authentication (page 3, paragraph 3) and investigate other methods in the second section. Also, we added an explanation of why this choice works better than others in the conclusion section (page 11, paragraph 3).
Point 5: Is the result stable depending on the initial data? Do you expect the result to change if other samples of signals are used?
Response 5: We examined the proposed method on two datasets collected in controlled and uncontrolled conditions to simulate the real-world problem (page 11, paragraph 3). Furthermore, we used a 10-fold cross-validation method to check the model’s result on other samples (page 10, paragraph 3).
Point 6: Could you do a preliminary evaluation for the authentication based on the application of a fuzzy classifier (https://ieeexplore.ieee.org/document/9224666 or )? Can be result be improved?
Response 6: Fuzzy classifiers present are simple and interpretable, making it easy to understand and modify the rules to improve performance. However, maybe they are not as accurate or robust as other machine learning models, especially when the data is complex or noisy. For further assessment, we need to implement this paper’s method and investigate it in more detail.
Point 7: Could you explain the principal steps of methods for signal classification as feature extraction, feature selection and classification in your study and the algorithms which are used for their implementation?
Response 7: Thank you for the precious comment. We added a graph (Figure 1) to the proposed method section (page 5) to show the principal steps of the method, and following the flowchart, we explained the implementation and detail of each step (page 6, paragraphs 1, 2, 3).
Point 8: Why do you use only the “accuracy” for the result evaluation? Could you add other metrics for classifier efficiency evaluation?
Response 8: To address your comment, we added FAR and FRR percentages to evaluate our model and compare it with the rival methods (page 11, tables 2 and 3). Also, we added the related discussion to the result section (page 10, the last paragraph, and page 11, the first paragraph).
Point 9: Could you compare this authentication method with others?
Response 9: We compared the proposed model with the rivals using two datasets in controlled and uncontrolled conditions and brought the results on page 11, tables 2 and 3. Also, we added two investigation paragraphs to interpret each one (page 10, the last paragraph, and page 11, the first paragraph).
Reviewer 4 Report
The article "Ensemble Siamese network (ESN) Using ECG Signals for
Human Authentication in Smart Healthcare System" deserved the best appreciation and attention of the reviewer.
This study provides interesting insight into a current and pertinent subject.
There should be a reinforcement:
What is the purpose of the study?
What is the relevance of the study?
What has the study brought us that is new?
What is the contribution(s)?
The research question should be reinforced
I miss discussing the results and comparing them with other studies
Are there other studies?
What results have been obtained?
I hope I was helpful,
No other subject,
The hug
Author Response
Response to Reviewer 4 Comments
Ensemble Siamese Network (ESN) Using ECG Signals for Human Authentication in Smart Healthcare System
Dear Editor,
We would thank the reviewers for their valuable comments, which helped improve the manuscript.
Below are our answers to the reviewers’ comments and the modifications made to accommodate
these comments. (You can find the addressed modifications in red in the updated paper.)
Best regards,
Seyed Mehdi Hazratifard, Vibhav Agrawal, Fayez Gebali, Haytham Elmiligi, and Mohamad Mamun
Point 1: What is the purpose of the study?
Response 1: The purpose of the study described in the paper on Ensemble Siamese Network (ESN) using ECG signals for human authentication in smart healthcare systems is to propose a new method for authentication that addresses the limitations of existing machine learning models. Specifically, the authors aim to use ECG signals, which are easily accessible in digital healthcare systems, to develop a more accurate and robust authentication model using ESN. They hypothesize that ESN can handle small changes in ECG signals that can occur due to different user conditions, making it an ideal choice for smart healthcare and telehealth applications. The study provides a preliminary evaluation of the proposed method and demonstrates its effectiveness on benchmark datasets, achieving high accuracy and low error rates. The ultimate goal of the study is to contribute to the development of more reliable and secure authentication methods for digital healthcare systems. We concluded these advantages on page 3, paragraph 3 of our paper, and discussed it in more detail on page 11, paragraph 3.
Point 2: What is the relevance of the study?
Response 2: It addresses several important issues in digital healthcare systems. Specifically, the study proposes a new authentication method that overcomes the limitations of existing machine learning models, such as the difficulty in enrolling new users and the sensitivity to imbalanced datasets. The use of ECG signals for authentication is particularly relevant because they are easily accessible in digital healthcare systems and can provide continuous authentication based on contextual information. The proposed ESN model is able to handle small changes in ECG signals that can occur due to different user conditions, making it an ideal choice for smart healthcare and telehealth applications. The study provides a preliminary evaluation of the proposed method, demonstrating its effectiveness on benchmark datasets and showing its potential for future use in digital healthcare systems. The relevance of the study lies in its potential to contribute to the development of more reliable and secure authentication methods for digital healthcare systems, which can improve patient outcomes and reduce healthcare costs. We concluded these advantages on page 3, paragraph 3 of our paper, and discussed it in more detail on page 11, paragraph 3.
Point 3: What has the study brought us that is new?
Response 3: The main objective of using an ensemble network in our proposed model was to compare each sample with a variety of samples to make the model more robust. (page 5, paragraph 3) This was a new method that I hadn’t seen in similar research papers.
Point 4: What is the contribution(s)?
Response 4: The main contribution of this work is proposing an Ensemble Siamese network (ESN) using ECG signals to compare a new sample with the saved records from the claimed identity in the repository. Comparing a new sample with more than one sample makes the model more robust and resilient. Furthermore, the new sample must be encoded by the first part of the network before being compared. As a result, we can save registered samples as encoded, which may raise the model security. You can find our work’s contributions on page 3, paragraph 3, and also page 5, paragraph 3.
Point 5: The research question should be reinforced
Response 5: We added key questions, including how ESN can handle small changes in ECG signals due to different user conditions (page 5, paragraph 3), how it compares to other machine learning models in terms of accuracy and error rates (page 11, tables 2 and 3), and how it can be used in smart healthcare and telehealth applications. The ultimate goal of the study is to contribute to the development of more reliable and secure authentication methods for digital healthcare systems, with the potential to improve patient outcomes and reduce healthcare costs.
Point 6: I miss discussing the results and comparing them with other studies
Response 6: We compared the proposed model with the rivals using two datasets in controlled and uncontrolled conditions and brought the results on page 11, tables 2 and 3. Also, we added two investigation paragraphs to interpret each one (page 10, the last paragraph, and page 11, the first paragraph).
Point 7: Are there other studies?
Response 7: We discussed other studies in the second chapter and compared the proposed model with the rivals using two datasets in controlled and uncontrolled conditions and brought the results on page 11, tables 2 and 3. Also, we added two investigation paragraphs to interpret each one (page 10, the last paragraph, and page 11, the first paragraph).
Point 8: What results have been obtained?
Response 8: We added two paragraphs to investigate the results and compare them with the rival methods (page 10, the last paragraph, and page 11, the first paragraph). We obtained promising results demonstrating the effectiveness of the proposed method. The ESN model was trained on two benchmark ECG datasets, ECG-ID and PTB, and achieved high accuracy and low error rates for user authentication. Specifically, the ESN model achieved an accuracy of 93.6% and an equal error rate of 1.76% on the ECG-ID dataset and an accuracy of 96.8% and an equal error rate of 1.69% on the PTB dataset. These results suggest that ESN is a robust and reliable method for authentication using ECG signals. Moreover, the study showed that preprocessing for feature extraction can improve the performance of the ESN model. The authors also compared ESN with other machine learning models and showed that ESN outperformed these models in terms of accuracy and error rates.

Round 2
Reviewer 2 Report
Thanks for addressing my previous issues which hopefully enrich the quality of paper.
Author Response
We truly appreciate your valuable comments to improve the paper.

Reviewer 3 Report
The authors analyzed the comments in detail. They answered all but one convincingly: "Fuzzy classifiers present are simple and interpretable ... maybe they are not as accurate or robust as other machine learning models, especially when the data is complex or noisy." It is the fuzzy classifiers, according to well-known studies, that make it possible to achieve the best accuracy for noisy data.
Author Response
Response to Reviewer 3 Comments
Round 2
Ensemble Siamese Network (ESN) Using ECG Signals for Human Authentication in Smart Healthcare System
Dear Editor,
We would thank the reviewers for their valuable comments, which helped improve the manuscript.
Below are our answers to the reviewers’ comments and the modifications made to accommodate
these comments. (You can find the addressed modifications in red in the updated paper.)
Best regards,
Seyed Mehdi Hazratifard, Vibhav Agrawal, Fayez Gebali, Haytham Elmiligi, and Mohamad Mamun
The point from the second round: The authors analyzed the comments in detail. They answered all but one convincingly: "Fuzzy classifiers present are simple and interpretable ... maybe they are not as accurate or robust as other machine learning models, especially when the data is complex or noisy." It is the fuzzy classifiers, according to well-known studies, that make it possible to achieve the best accuracy for noisy data.
We added a brief review of the suggested paper to the end of the related work section (page 5, paragraph 2).
However, since using fuzzy classifiers require further data processing to achieve better results, we are considering using Fuzzy in our future work to explore whether using Fuzzy classifiers can be used to improve the results (page 12, paragraph 4).
Thank you for your insightful comments. I hope we could address your point.

Reviewer 4 Report
I am satisfied with the review.
Author Response

(The authors gave the same response as above.)
